# Research on Braking Efficiency of Master-Slave Electro-Hydraulic Hybrid Electric Vehicle

**Junyi Wang** [1,2], **Tiezhu Zhang** [1,2], **Hongxin Zhang** [1,2,*], **Jian Yang** [1,2], **Zhen Zhang** [1,2] and **Zewen Meng** [1,2]

1   College of Mechanical and Electrical Engineering, Qingdao University, Qingdao 266071, China; 17861403371@163.com (J.W.); zhangtz@sdut.edu.cn (T.Z.); yangxiaoming8533@163.com (J.Y.); rexzz9916@163.com (Z.Z.); mengzewen2020@163.com (Z.M.)
2   Power Integration and Energy Storage Systems Engineering Technology Center (Qingdao), Qingdao 266071, China
*   Correspondence: qduzhx@126.com; Tel.: +86-13573865229

**Abstract:** To address the problems of short-rangee and poor braking safety of electric vehicles, this paper proposes a master-slave electro-hydraulic hybrid passenger car drive system based on planetary gear. The system couples the electrical energy output from the electric motor with the hydraulic energy output from the electro-hydraulic pump/motor through the planetary gear. The hydraulic system is used as the auxiliary power source of the power system giving full play to the advantages of the hydraulic system and the electric system. After theoretical analysis, this paper establishes a master-slave electro-hydraulic hybrid electric vehicle (MSEHH-EV) model based on planetary gear in AMESim software. A braking energy recovery control strategy is designed with the maximum braking energy recovery efficiency as the target. Braking strength determines the switching of braking modes. Finally, comparing the certified pure electric vehicle (EV) model in AMESim, we are able to substantiate the superiority of the strategy proposed in this paper. The simulation results revealed that the battery consumption rate of the new power vehicle is reduced by 17.766%, 11.358%, and 9.427% under UDDS, NEDC, and WLTC conditions, respectively, which supports the range. At the same time, the braking distance is significantly shortened, and the maximum braking distance is shortened by 15.65 m, 21.97 m, and 21.45 m, respectively, under the three operating conditions, which improves the braking safety.

**Keywords:** electro-hydraulic hybrid; braking research; control strategy; energy recovery; braking safety



## 1. Introduction

### 1.1. Current Status of Research

Against the backdrop of the stagnant development of EV technology, the auto industry has also accelerated the pace of new energy vehicle development and promotion. The environmental sustainability stimulates the development of EVs with great energy saving and emission reduction effects [1]. EV has a broad development prospect, but at present, there are still obvious shortcomings. As we all know, due to the small battery storage capacity and low use rate of brake energy recovery, the range of EV is short and they cannot fully meet the energy consumption demand of EVs [2]. Furthermore, the frequent charging and discharging will also speed up the speed of battery aging. Battery systems are crucial to the safety and durability of EVs [3]. Hong first applied the Long Short-Term Memory (LSTM) to voltage prediction and fault prognosis of the battery system [4]. Hong proposed a new dynamic drive system, the load-isolated EV, and designed the electric system parameters to matched it. He verified the feasibility and superiority of the load-isolated EV system through performance simulation [5]. Wang established a battery-flywheel composite energy storage system. Compared with a single battery system, the recovery energy of this method increased by 1.17 times and the greatest charging current of

the battery decreased by 42.27%. Also, the braking distance of pure EVs still has a lot of room for improvement, and braking safety is a hidden problem [6].

The hybrid combination could be the forword-looking technologies that supports the development of EVs in the modern development of the Modern Automobile [7]. The advent of the Hybrid Electric Vehicle (HEV) has ameliorated many of the problems associated with EVs. Yang designed a new electro-hydraulic hybrid drivetrain to avoid the impact of excessive charging and discharging currents on the battery cycle life during driving or braking of pure EVs, which significantly improved the vehicle economy [8]. Honey models an electro-hydraulic hybrid vehicle and compares it to an EV under urban driving conditions. The results show that the method not only reduces the energy consumption of the battery but also reduces the transient current of the battery. He demonstrated the feasibility of electric-hydraulic hybrid systems for passenger cars [9]. Hwang designed a hydraulic electric hybrid vehicle (HEHV) for medium and heavy-duty vehicle applications, and based on the simulation, he designed an energy management strategy. The simulation results show that the designed HEV has improved electric economy and energy performance compared with the EV, which fully validates the advantages of HEV [10].

HEVs are mostly oil-electric hybrid, and there are efficiency problems with energy conversion and power shunt. To achieve better energy conversion and better energy efficiency, Hong proposed a new MEHPC EV with electric peak torque as the optimization target, he also proposed the distribution method of electro-hydraulic ratio under different operating conditions to improve the battery SOC [11]. Yang proposed a new electromechanical-hydraulic dynamic coupling transmission system (MEH-DCDS), the addition ofhydraulic system gives play to its advantages of high torque peculiarities. It combines a motor with a swashplate hydraulic pump to achieve the mutual conversion of mechanical, hydraulic, and electrical energy. The simulation proves that the system has better acceleration and lower power consumption [12]. Borhan to enhance the fuel efficiency of hybrid vehicles and design a near-optimal energy management strategy, gave the simulation results for a closed-loop high-fidelity model of a power-sharing HEV with many standard drive cycles and different controllers. Significant improvements in fuel economy are achieved compared to existing commercial controllers [13]. Hong investigates motor torque control algorithms for power-split hybrid vehicles to optimize engine operation. He designed the control algorithm to control the two motors, the engine power and battery power depending on the driver's intention to accelerate, so that the engine runs on the optimal operating line. He designed a dual planetary gear power-split hybrid car. The simulation results a surface that the engine can run on the optimal operating line while meeting the battery energy requirements [14].

In order to improve the performance of HEV and accelerate the industrialization of HEV, the energy management technology of HEV has been extensively and deeply studied [15]. Yang established an energy management strategy that reduces the time it takes for a vehicle to reach start speed and significantly reduces the motor peak torque [16]. Bedir proposed an automatic management strategy that made a 50% improvement in the driving range of the Honda insight [17]. Deng proposed the adaptive strategy of least equal fuel consumption by analyzing the shortcomings of the existing HEV energy management strategy and expounded on the future development direction of Chinaese HEV energy management strategy [18]. Zhang investigated adaptive equal consumption minimization strategy (A-ECMS) for parallel plug-in HVs based on real-time road condition prediction and driving behavior recognition, which has the potential to achieve better fuel economy. The improvement in the braking energy recovery efficiency of hybrid vehicles cannot be achieved without a reasonable energy recovery control strategy. As a result of the improvement in energy recovery efficiency, hybrid vehicles have greatly improved their range [19]. Hong, in order to achieve fast SOC balance and adaptive recovery of braking energy, was proposed a discrete input-output battery grouping system based on the overturned four-quadrant topology structure [20]. Geng proposed a two-layer multi-parameter hybrid vehicle braking energy recovery control strategy, which can integrate

the motor braking system with the mechanical braking system, so as to achieve better braking effect while ensuring adequate braking performance and safety performance of the vehicle [21]. Jiang proposed an energy management strategy, it makes the pure electric bus reach the maximum braking energy recovery efficiency. At the same time, the friction braking force of front and rear wheels can be distributed in a fixed proportion to ensure the maximum braking efficiency and braking stability [22]. Zhang developed a new type of vehicle that avoids the rear wheel locking before the front wheel and determines the distribution ratio of braking force in each case. He designed a control strategy based on maximum braking energy recovery efficiency and verified it through road tests. The simulation results show that the optimal energy recovery control strategy is very effective for improving the braking energy recovery efficiency of HEVs [23]. Wang conducted a theoretical analysis of the braking force distribution, calculated the range of application of the braking force distribution law, and proposed a braking energy recovery greatest control strategy to divide the greatest braking force to the motor in the range specified by regulation No.13 of automobile Standards issued by the Economic Commission of Europe (ECE), the energy recovery effect of feedforward braking is better than RWD, and the maximum braking energy recovery control strategy proposed is effective and closest to the maximum potential energy of feedforward braking energy recovery control strategy [24]. Deng calculated the joint battery/motor/CVT efficiency and proposed a braking energy control strategy. Compared with the previous joint battery/motor efficiency control strategy, the braking energy recovery rate was improved by 4.09% [25]. Meng proposed a new electromechanical-hydraulic coupled EV and designed a matching energy management strategy. According to the different braking speeds of the vehicle, the conversion of energy recovery mode is optimized. Finally, the feasibility of the proposed control strategy is verified by simulation of AMESim and Simulink [26].

Planetary gears have the advantages of compact structure, small size, and light weight, and are often used as power coupling elements in multi-source hybrid power systems. Yin proposed a novel planetary gear hybrid power system. It combines EV, and HEV into one vehicle. The regeneration during braking optimizes the fuel economy, and it is simple to configure, easy to assemble, and flexible in application [27].

HEVs have better braking stability and braking efficiency. Ji establishes an electro-hydraulic hybrid braking system simulation platform based on CARSIM and MATLAB/Simulink and puts forward the corresponding braking control strategy. Hybrid braking and hydraulic braking correspond to vehicle braking with different adhesion coefficients. The simulation results a surfacethat when the EV is in a low braking intensity condition, it can not only recover the braking energy but also meet the braking performance requirements [28].

Based on the above research, this paper proposes the MSEHH-EV based on planetary gear, designs a braking energy recovery control strategy to match it, and conducts a study on braking efficiency, which is of great significance for future research on HEVs.

### 1.2. Contribution of This Paper

1. A novel MSEHH-EV based on the planetary gear is proposed;
2. A braking energy recovery control strategy matching the MSEHH-EV is designed, which is capable of braking mode switching by threshold setting;
3. A joint simulation with AMESim and Simulink verifies the correctness of the vehicle and the feasibility of the corresponding braking energy recovery control strategy, and the simulation results showed that the model vehicle had better braking performance than EVs;

### 1.3. Article Structure

The sections of this paper are structured as follows: Section 2 introduces the system composition and operating principle of this vehicle. Section 3 analyzes the three braking modes of the vehicle. Section 4 is the theoretical modeling and strategy design. Section 5 is the modeling and simulation. Section 6 is the conclusion and outlook.

## 2. System Composition and Working Principle

### 2.1. Composition of the System

Figure 1 illustrates the structure of the MSEHH-EV system based on planetary gear. The vehicle can realize the power coupling of mechanical energy, electric energy, and hydraulic energy. Through the synergy and coupling of electromechanical and hydraulic power, it realizes the efficient regenerative use of braking energy, fast starting and braking, motor working in the efficient working condition zone, synchronization of vehicle travel and hydraulic load work, recycling of inertial energy of hydraulic working device, reduction of peak power of the motor, and load fluctuation without large current shock, etc. The performance of the vehicle is greatly improved by the compact power transmission structure and method.

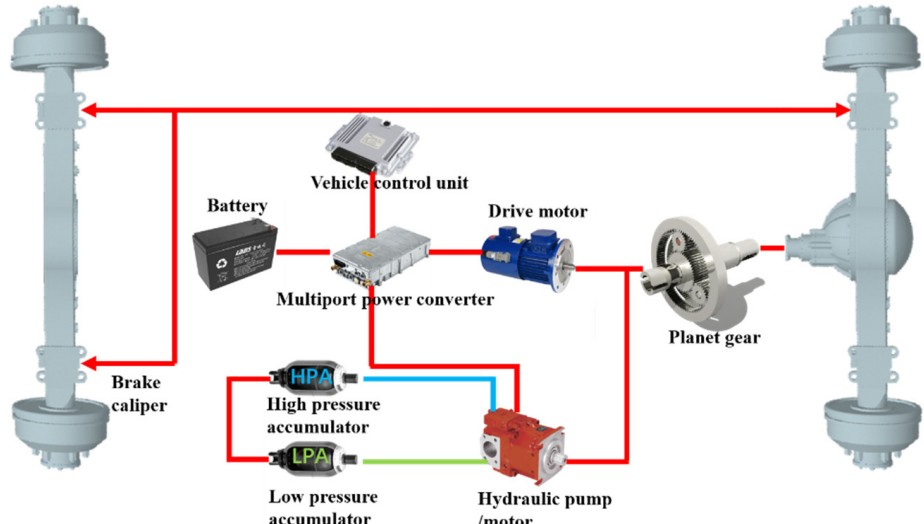

**Figure 1.** Structure of MSEHH-EV system based on planetary gear.

The system mainly includes a power battery, multi-port power converter, electrical equipment, motor, clutch brake, planetary row, clutch brake, drive axle, drive wheel, high-pressure accumulator (HPA) and low-pressure accumulator (LPA), electronically controlled hydraulic pump/motor, clutch brake, and other components.

### 2.2. Working Principle of Planetary Gear

The MSEHH-EV system enables the electric energy output from the electric motor and the hydraulic energy output from the electro-hydraulic pump/motor to be power coupled at the planetary row. The structure of the planetary gear used in this system is shown in Figure 2. The planetary gear consists of a ring gear, a planet gear, a sun gear, and a planet carrier.

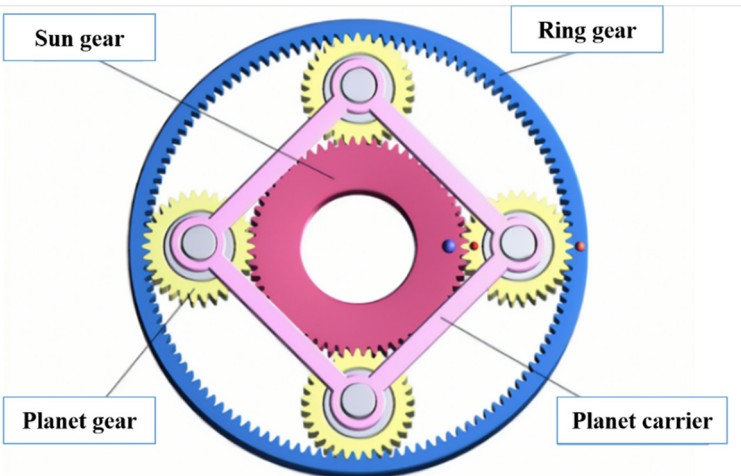

**Figure 2.** Structure of the planetary gear.

This system uses an input shaft connected to the sun gear, a planet carrier connected to the output shaft, and a ring gear meshed with the hydraulic power direct gear.

In the planetary gear mechanism, the speed ratio of the ring gear to the sun gear is taken as its basic characteristic parameter:

$$k_{\mathrm{p}} = \frac{z_2}{z_1} \tag{1}$$

where $k_{\mathrm{p}}$ is the characteristic parameter of the planetary gear; $z_2$ is the number of teeth of the ring gear; $z_1$ is the number of teeth of the sun gear.

The relationship between planet carrier, sun gear, and ring gear as follows:

$$n_3 = \frac{n_1}{(k_{\mathrm{p}} + 1)} + \frac{k_{\mathrm{p}} * n_2}{(1 + k_{\mathrm{p}})} \tag{2}$$

where $n_3$ is the speed of planet carrier, $n_1$ is the speed of sun gear, and $n_2$ is the speed of ring gear.

The torque relationship between the sun gear, planet carrier, and ring gear as follows:

$$T_1 : T_2 : T_3 = \frac{1}{k_{\mathrm{p}} + 1} : 1 : \frac{k_{\mathrm{p}}}{k_{\mathrm{p}} + 1} \tag{3}$$

where $T_1$ is the torque of sun gear, $T_2$ is the torque of ring gear, and $T_3$ is the torque of planet carrier.

## 3. Braking Mode Analysis

The MSEHH-EV has three braking modes, which are Mechanical Braking mode (MB) and Regenerative Energy Braking mode (REB), where REB is divided into Hydraulic Energy Regenerative Braking mode (HERB) and Electrical Energy Regenerative Braking mode (EERB).

At lower speeds, as shown in Figure 3a, the system enters the HERB mode, which uses the hydraulic system for braking energy recovery. The hydraulic pump/motor works in the hydraulic pump operating state, and the kinetic energy transfers through the output axle to the planet carrier, and then to the ring gear, which drives the hydraulic power direct gear to drive the hydraulic pump to rotate and pump the hydraulic oil in the LPA to the HPA to realize the conversion of mechanical energy to hydraulic energy. Hydraulic energy is converted from the accumulator to mechanical energy by the hydraulic pump/motor through the planetary gear. The hydraulic energy can provide the necessary power for

vehicle starting or acceleration reduces the impact of high current on the power battery during vehicle starting and acceleration and extends the service life of the power battery.

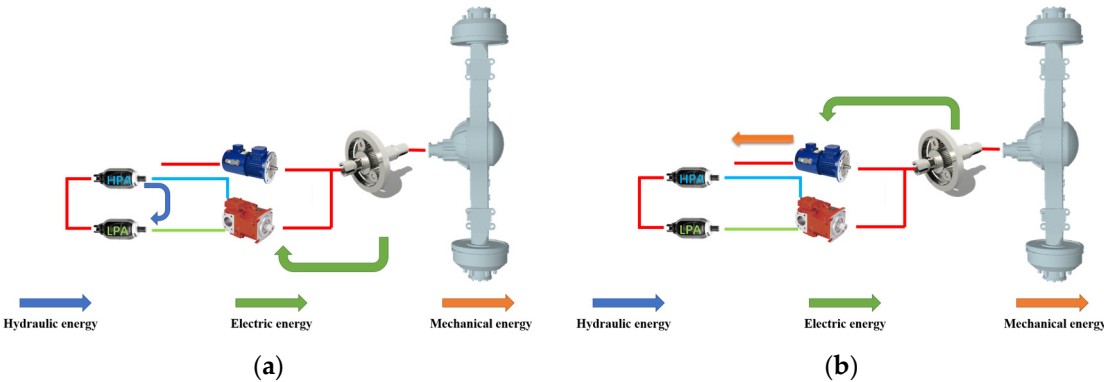

**Figure 3.** Two regenerative braking modes (**a**) HERB; (**b**) EERB.

At higher speeds, as shown in Figure 3b, it enters the EERB mode, and the working state of the motor is a generator. The kinetic energy on the axle goes through the output shaft to the planet carrier and then to the sun gear, and the drive motor connected to the sun gear plays the role of a generator and outputs electrical energy is recycled into the battery. If the battery capacity is full at higher speeds, Then the hydraulic system performs energy recovery. When driving, this energy can be reused to improve energy-saving effects. And when the vehicle is parked or running, the electrical energy in the power battery pack can be transformed into hydraulic energy and stored in the hydraulic accumulator.

Besides, to ensure safety under emergency braking conditions, it enters MB mode and only mechanical braking is performed.

## 4. Theoretical Modeling and Strategy Design

### 4.1. Model Establishment

To achieve the maximum braking energy recovery efficiency of the MSEHH-EV, this section provides a theoretical analysis of its main energy storage components. This work lays the foundation for subsequent research on braking energy recovery control strategies.

#### 4.1.1. Electric Energy Module Modeling

Electric energy is the main power source for EVs, so the establishment of the motor model is crucial.

When the vehicle is fully loaded and in pure EV mode on a smooth road, the power required to drive fast at top speed is:

$$P_e = \frac{1}{\eta_T}\left(\frac{Gfu_{\text{max}m}}{3600} + \frac{C_D Au_{\text{max}m}^3}{76,140}\right) + P_{\text{acc}} \tag{4}$$

where $G$ is the vehicle gravity, $f$ is the rolling resistance coefficient, $u_{\text{max}m}$ is the maximum speed in pure electric drive mode, $C_D$ is the air resistance coefficient, $A$ is the windward area of the electric vehicle, $P_{\text{acc}}$ is the total power consumed by the components; $\eta_T$ is the mechanical transmission efficiency.

In the electric drive mode, the power required for climbing at maximum speed is determined according to the actual driving conditions of the vehicle and the efficiency of the transmission system:

$$P_c = \frac{1}{\eta_T}\left(\frac{Gf\cos\theta u_1}{3600} + \frac{G\sin\theta u_1}{3600} + \frac{C_D Au_1^3}{76,140}\right) + P_{\text{acc}} \tag{5}$$

where $\theta$ is the climbing gradient and $u_1$ is the speed at the maximum climbing gradient.

In full-electric drive mode, the power required for acceleration is:

$$P_{\text{maxa}} = \frac{1}{\eta_{\text{T}}} \left( \frac{Gfu_2}{5400} + \frac{\delta mu_2^2}{3600 \times 7.2 \times \text{t}} + \frac{C_{\text{D}}Au_2^3}{76,140 \times 2.5} \right) + P_{\text{acc}} \tag{6}$$

where $m$ is the total mass of the vehicle and $u_2$ is the end speed of acceleration from the original starting point.

The peak power $P_{\text{max}}$ of the motor meets the requirements of acceleration, hill climbing, maximum speed driving, and other working conditions.

This paper selects the peak power of motor $P_{\text{max}}$ = 30 kW, rated power $P_{\text{m}}$ = 15 W, and the overload coefficient is 2.

For the power pack of the MSEHH-EV, the relationship between the peak output power and the peak motor power is as follows:

$$P_{\text{maxb}} = \frac{P_{\text{maxm}}}{\eta_b} \tag{7}$$

The peak torque of the motor is:

$$P_{\text{maxm}} \geq \max(P_{\text{c}}, P_{\text{e}}, P_{\text{maxa}}) \tag{8}$$

The Motor output voltage is:

$$U_{out} = U_{\text{oc}} - IR \tag{9}$$

where $U_{\text{oc}}$ is the open-circuit voltage of the power cell, $I$ is the current power cell electricity, $R$ is the internal resistance of the power cell.

4.1.2. Hydraulic Energy Module Modeling

When the vehicle is fully loaded and the vehicle is in hydraulic drive mode, the power output of the hydraulic pump/motor module required to speed up or start on a flat surface is:

$$P_{\text{maxa}} = \frac{1}{\eta_{\text{T}}} \left( \frac{Gfu_2}{5400} + \frac{\delta mu_2^2}{3600 \times 7.2 \times \text{t}} + \frac{C_{\text{D}}Au_2^3}{76,140 \times 2.5} \right) + P_{\text{acc}} \tag{10}$$

Maximum torque is achieved by hydraulic pump/motor $T$:

$$T = \frac{\Delta p V_p}{20\pi} \tag{11}$$

where $\Delta_P$ is the Pressure difference between the inlet and outlet the of hydraulic pump, and $V_p$ is the hydraulic pump displacement.

The maximum energy recovered by the hydraulic accumulator during braking is calculated as shown in the following equation:

$$E_{\text{pre}} = \frac{p_0 V_0}{n - 1} \left[ \left( \frac{p_2}{p_0} \right)^{\frac{n-1}{n}} - 1 \right] \tag{12}$$

where $E_{\text{pre}}$ is the maximum energy recovered by the hydraulic accumulator, $p_0$ is the precharge pressure value of hydraulic accumulator, $P_2$ is the maximum working pressure of the hydraulic accumulator, $V_0$ corresponds to the volume of the gas when the pressure is $P_0$, and $n$ is the multivariable index of the gas, 1 in isothermal condition and 1.4 in adiabatic condition.

Finally, the maximum pressure of 35 MPa for the HPA, 18 MPa for the LPA, and 10 MPa for the minimum pressure are selected as the parameter settings.

The following relationship exists between $p_0$ and $p_2$.

$$P_0 V_0 = P_2 V_2 \tag{13}$$

The hydraulic system brake energy recovery rate $\eta$ is:

$$\eta = \frac{E_{\text{pre}}}{E} \tag{14}$$

$E$ stands for the kinetic energy of the car:

$$E = \frac{1}{2} M \left( v_2^2 - v_1^2 \right) \tag{15}$$

where $M$ stands for mass of the vehicle, $v_1$ is the speed of the vehicle at the initial moment, and $v_2$ is the speed of the vehicle at the final moment.

### 4.1.3. Other Basic Parameters

Table 1 is the other basic parameters of the vehicle.

**Table 1.** Other basic parameters of the vehicle.

| Parameters | Value |
|---|---|
| Overall vehicle mass | 1026 kg |
| Maximum speed | 150 m/s |
| Windward area | 2.26 m$^2$ |
| Power battery voltage | 310 V |
| Hydraulic pump/motor displacement | 100 cc/rev |
| Transmission efficiency | 0.85 |

### 4.2. Braking Force Distribution

While the MSEHH-EV is braking, the front axle is only in MB, and the axle of the rear axle is subject to hydraulic regenerative braking force, electrical regenerative braking force, and mechanical braking force. And the distribution of each braking force is shown in Figure 4. It is clearly stipulated in regulation No.13 of the automobile Standards issued by the Economic Commission of Europe (ECE):Braking force distribution should meet the requirements of vehicle safety and operational stability. The addition of a hydraulic system will change the distribution of brake torque, so the regenerative braking force distribution strategy needs to be designed to meet the demands of ECE regulations.

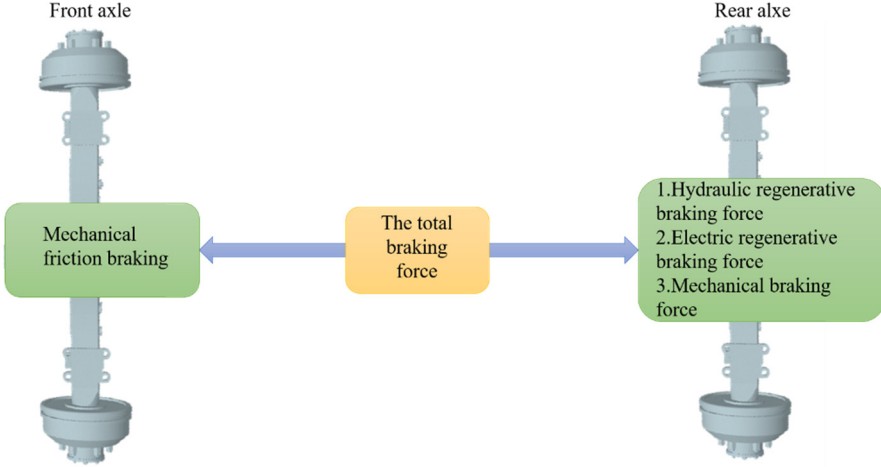

**Figure 4.** Braking force distribution relationship diagram.

(1)    Under small braking intensity: $Z \leq 0.1$

Braking intensity is low, now ECE regulations do not have strict restrictions on the braking force apportionment coefficient, to get a better braking energy recovery effect, the design of the rear axle regenerative braking provides all the braking force, and the front and rear axle MB force are 0.

$$F_{bf} = 0 \tag{16}$$

$$F_{br} = Mgz \tag{17}$$

where $F_{bf}$ stands for front braking force, $F_{br}$ stands for rear braking force, $g$ stands for the acceleration of gravity, and $z$ stands for braking force strength.

The front and rear axle braking ratios are distributed as follows:

$$F_{bf-fric} = 0 \tag{18}$$

$$F_{reg} = Mgz \tag{19}$$

where $F_{bf\text{-}fric}$ stands for front axle mechanical braking force, and $F_{reg}$ stands for the regenerative braking force of the motor.

(2)    Under medium braking intensity: $0.1 < Z < 0.7$

When the vehicle is under medium braking intensity, the front and rear axle braking ratios are distributed as follows:

$$F_{bf-fric} = F_{req} - F_{reg} \tag{20}$$

$$F_{br-fric} = 0 \tag{21}$$

$$F_{reg} = \varphi \left( \frac{Ga}{L} - \frac{F_{req}h_g}{L} \right) \tag{22}$$

where $F_{req}$ stands for demand braking force; $F_{br\text{-}fric}$ stands for rear axle mechanical braking force, $\varphi$ stands for coefficient of adhesion; $G$ stands for the gravity of the car; $a$ stands for distance from the center of mass of the car to the front axle; $h_g$ stands for the height of the center of mass of the car, and $L$ stands for the wheelbase of the vehicles.

(3)    Under emergency braking strength: $Z \geq 0.7$

Under the emergency braking condition, in order to ensure the braking safety and stability of EVs, regenerative braking is not involved, and the braking power demand of the EV is provided by the mechanical braking power of the brakes for both axles.

$$F_{bf-fric} = \varphi \frac{G}{L} \left( b + zh_g \right) \tag{23}$$

$$F_{br-fric} = \varphi \frac{G}{L} \left( a - zh_g \right) \tag{24}$$

According to the proposed braking power distribution method, the logic diagram of regenerative braking system control strategy is established, as shown in Figure 5: In the diagram, "Y" and "N" indicate "compliance" and "non-compliance", respectively.

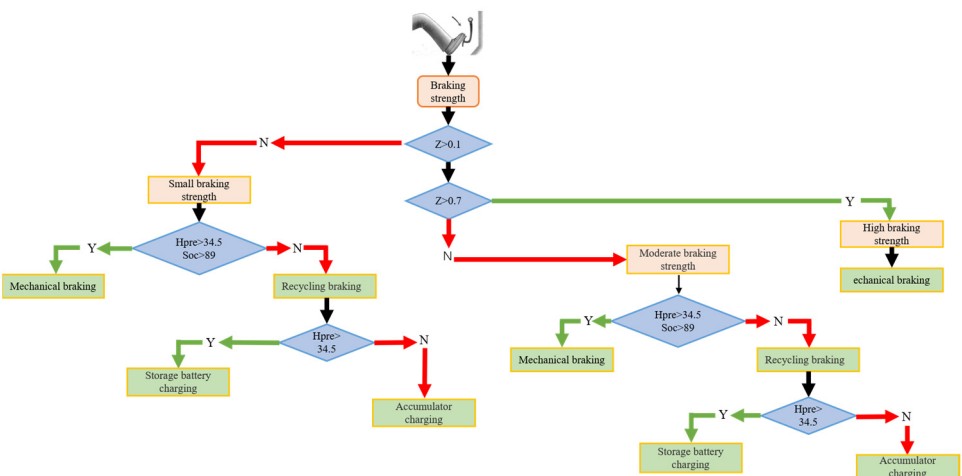

**Figure 5.** Regenerative braking control strategy logic.

Where $Z$ is the braking intensity signal, $H_{pre}$ is the high-pressure accumulator pressure, and SOC is the current state of charge signal of the power battery.

### 4.3. Control Strategy Design

### 4.3.1. Control Strategy Logic

As shown in Figure 6, the operating condition mode of a car is firstly divided into parking mode, driving mode, and braking mode. The drive mode is divided into hydraulic drive mode, pure electric drive mode, and electro-hydraulic hybrid drive mode; The braking mode is divided into MB, and REB. Among them, the REB is divided into ERB, and HRB.

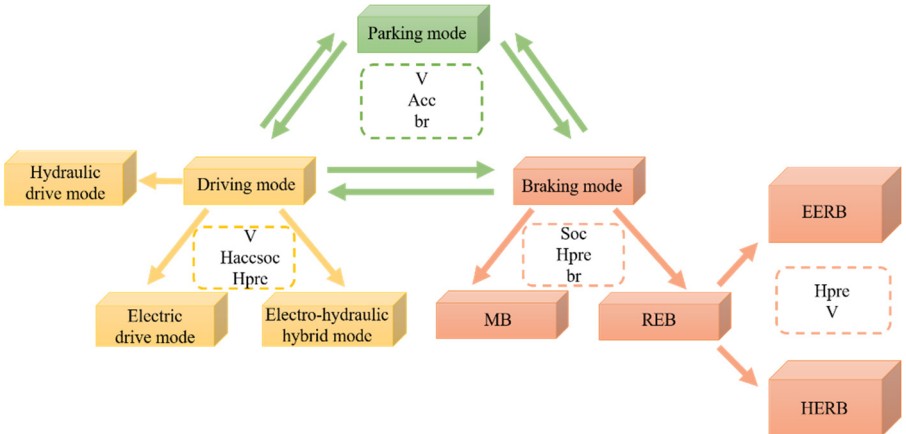

**Figure 6.** Operating modes of MSEHH-EV.

In the basic three operating modes, the operating modes are switched on the basis of the vehicle speed signal $V$, acceleration signal $A_{cc}$, and brake signal $b_r$. In the drive mode, the switchover is based on the vehicle speed signal V, the HPA SOC signal $H_{accSOC}$, and the high and low-pressure accumulator pressure difference $H_{Pre-LPre}$. In the braking mode, firstly, the MB and REB are converted according to the battery SOC signal, high-pressure accumulator pressure $H_{pre}$ signal, and $b_r$ signal. Secondly in the REB, the EERB and HERB are converted according to the V and $H_{accSOC}$ signal.

### 4.3.2. Implementation of the Control Strategy

(1)    Stop mode: All output signals are 0.

(2) Hydraulic drive mode: all output signals are 0, the hydraulic pump/motor is in working state, and the SOC of the accumulator can be described as:

$$SOC = \frac{C_p - G_{pp}}{Max_p - G_{pp}} \tag{25}$$

where $C_p$ is Current Pressure; $G_{pp}$ is Gas pre-charge pressure; and $Max_p$ is Maximum pressure.

The hydraulic pump/motor displacement $V$ = 100 mL/r.

The torque calculation formula is:

$$T = \frac{\beta \cdot V \cdot \Delta P}{20 \cdot \pi} \tag{26}$$

where $\beta$ is the hydraulic pump swashplate opening $(-1, 1)$.

(3) Motor drive mode: Based on the received $A_{cc}$ from the driver module, the motor torque required under the current operating conditions is output by multiplying the motor's external characteristic curve by:

$$T \bullet Acc = T_m \tag{27}$$

where the torque at the current speed and voltage is $T$; The actual torque output signal of the motor is $T_m$, and the Unit is $N \bullet m$.

(4) Electro-hydraulic hybrid drive mode: The hydraulic motors and electric motors work together to power the vehicle. The hydraulic system and motor system distribute torque in a 1:1 electro-hydraulic ratio. Because torque is proportional to power. Therefore, if you control the power distribution, you control the torque distribution. The motor torque is the accelerator pedal signal multiplied by the motor external characteristic curve to get the demand torque under the current working speed, then, 50% torque allocate to the electric motor, and the swashplate opening of the hydraulic motor is calculate as follows:

$$\beta = \frac{T \bullet Acc \bullet 0.5 \bullet 20 \bullet \pi}{\Delta p \bullet 100} \tag{28}$$

In addition, to carry out a comparison between the torque that the hydraulic motor can provide and the demand torque (50% of the torque), when the torque of the hydraulic motor does not satisfy the requirement torque, the motor can provide insufficient torque, as shown in the following formula:

$$T \bullet Acc \bullet 0.5 + \left( T \bullet Acc \bullet 0.5 - \frac{\Delta P \bullet 100 \bullet \beta}{20 \bullet \pi} \right) = T_m \tag{29}$$

(5) MB: No output of motor torque and swashplate angle, the only output of $b_r$ signal. The current $b_r$ signal is multiplied by the maximum braking torque of the front axle to get the braking torque required for the current working condition and output to the vehicle brake:

$$T_b = 1000 \bullet br \tag{30}$$

where $T_b$ is the braking torque required to be provided by the vehicle brake.

(6) EERB: control motor torque to achieve the purpose of motor output negative torque recovery braking energy. The current brake pedal signal is multiplied by 1000 to get the required braking torque at the axle of the current operational states. The torque provided by the current motor can be obtained through the motor external characteristic curve. Compare the torque between them, if the torque provided by the motor is less than the required braking torque, the motor operates at the current torque that it can provide, and insufficient torque is compensated by friction braking

of the vehicle mechanical brake. When the torque provided by the motor is greater than or equal to the braking torque required, then output the motor torque of the braking torque currently required.

(7)   HERB: first calculate the braking torque required at the axle under current operating conditions. Then calculate the torque that can be delivered to the axle by the hydraulic pump/motor under the current pressure. If the torque provided by the hydraulic pump is less than the required braking torque, the hydraulic pump calculates the swashplate angle based on the current torque provided, and insufficient torque is compensated by friction braking of the vehicle brake. If the torque provided by the hydraulic pump is greater than the required braking torque, the hydraulic pump calculates the swashplate angle according to the braking torque required by the axle.

## 5. Modeling and Simulation

According to the theoretical knowledge mentioned earlier in this paper, an MSEHH-EV simulation model is established in AMESim, and a joint simulation is established with Simulink through the joint simulation interface. As shown in Figure 7, according to the driving conditions, the driver model reacts, the control signal is transmitted to the controller, the controller sends a signal to the electric motor or hydraulic pump/motor, and the electric motor and hydraulic pump/motor output torque are coupled at the planetary gear, and then transmitted to the axle through the clutch.

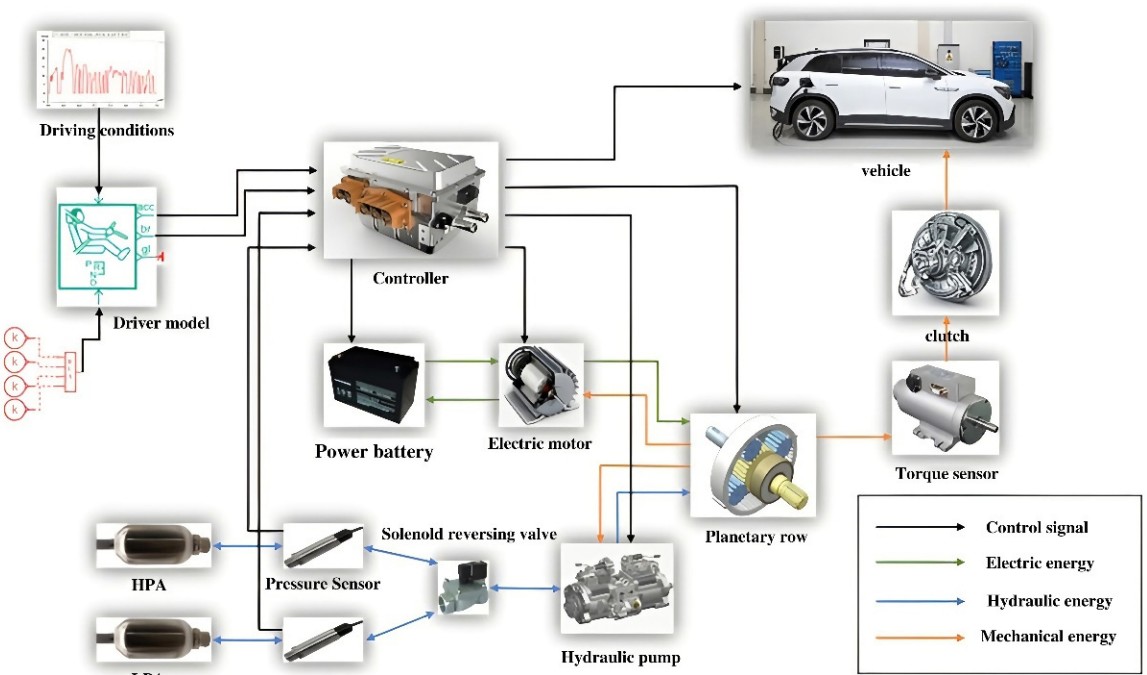

**Figure 7.** MSEHH-EV simulation model.

To verify the advantages of MSEHH-EV over pure EVs in terms of braking performance, simulations were conducted under three operating conditions: Urban Dynamometer Driving Schedule (UDDS), New European Driving Cycle (NEDC), and World Light Vehicle Test Procedure (WLTC). The three working conditions are described in Table 2.

**Table 2.** Introduction to the three working conditions.

| Main Features | UDDS | WLTC | NEDC |
|---|---|---|---|
| Running time/s | 1370 | 1800 | 1180 |
| Mileage/km | 12.07 | 23.27 | 11.007 |
| Max speed/(km/h) | 91.2 | 131.3 | 120 |
| Average speed/(m/s) | 31.5 | 46.54 | 33.68 |
| Component | 2 | 4 | 2 |
| Main features | UDDS | WLTC | NEDC |

Firstly, as shown in Figure 8, the MSEHH-EV was simulated under three operating conditions, and the braking distance was compared with that of the pure EV model certified by AMESim. It can be easily concluded from the speed curves (the curve above each figure) that the stopping time of MSEHH-EV is earlier than that of EV in each braking condition in the three conditions, and the maximum difference of braking distance between a MSEHH-EV and EV in the three conditions is 15.65 m, 21.97 m, and 21.45 m, respectively, calculated from the displacement curves, which fully verifies that the simulation model has better braking safety.

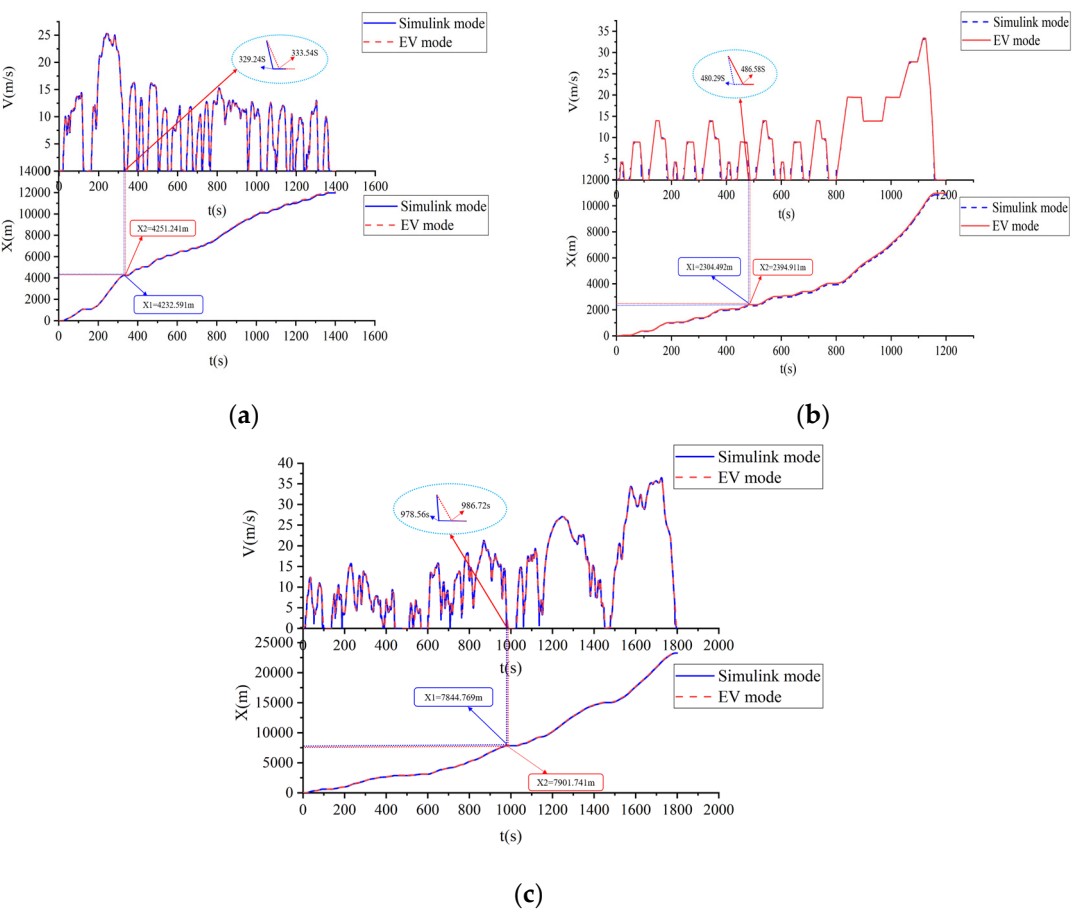

**Figure 8.** Comparison of braking distance between MSEHH-EV and EV (**a**) UDDS (**b**) NEDC (**c**) WLTC.

Figure 9 shows the change of battery SOC for both models. Among them, Figure 9a is the simulation result under the UDDS condition, Figure 9b is the simulation result under the NEDC condition, and Figure 9c is the simulation result under the WLTC condition. The blue line is the battery SOC curve of the test vehicle, and the red line is the battery SOC curve of the EV, and it is easy to conclude that the energy consumption of this simulation model is less than that of the EV under the three working conditions. The energy consumption is

reduced by 17.766% in UDDS operation, 11.358% in NEDC operation, and 9.427% in WLTC operation, as shown in Table 3, which proves that the model has a better electric energy recovery effect, and also proves that the incorporation of a hydraulic system in the model has a significant energy-saving effect. In summary, the MSEHH-EV has a better braking energy recovery effect, thus, in conclusion, the MSEHH-EV has a better braking energy recovery effect and thus a better range than the EV.

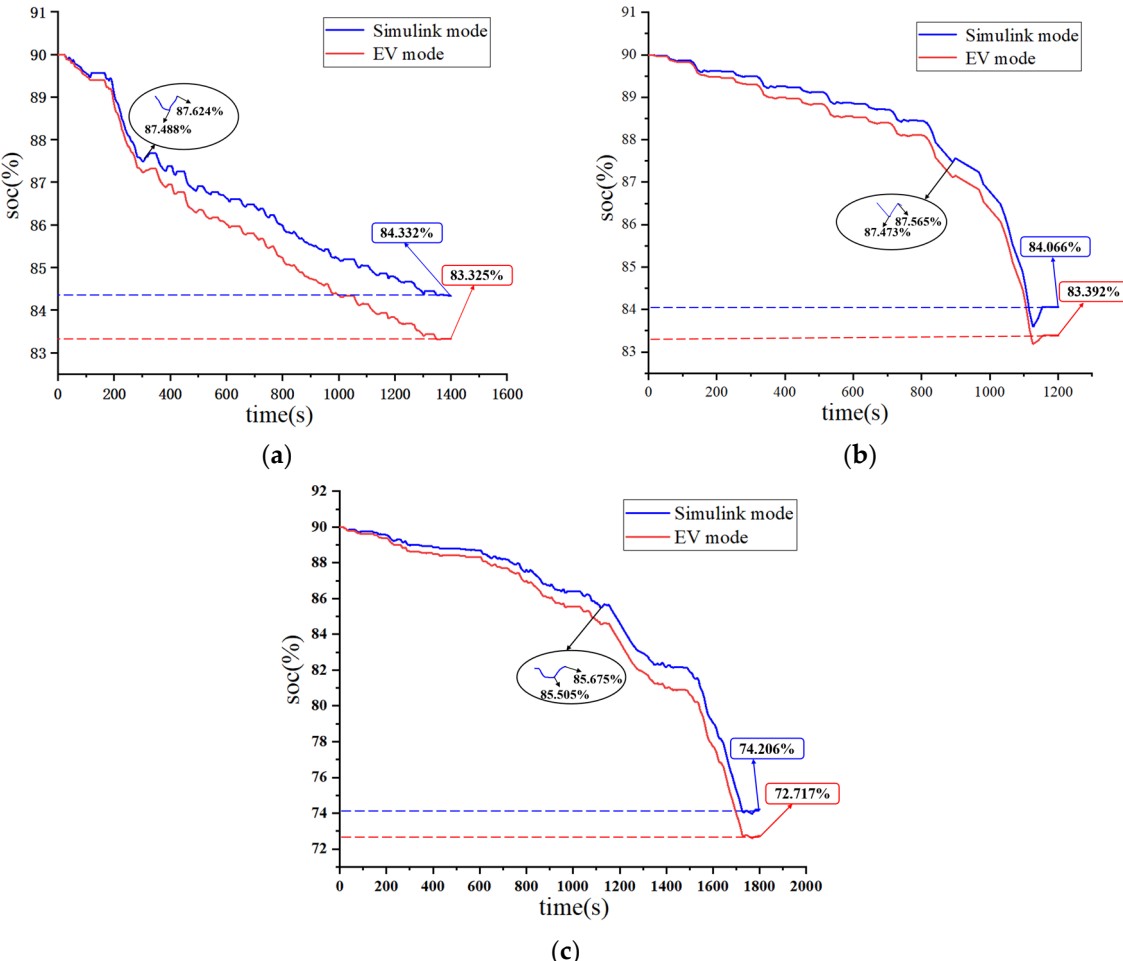

**Figure 9.** Battery SOC comparison chart (**a**) UDDS (**b**) NEDC, (**c**) WLTC.

**Table 3.** Simulation results of three working conditions.

| Operation Condition | Braking Distance Reduction | Reduce Energy Consumption |
|---|---|---|
| UDDS | 15.65 m | 17.766% |
| NEDC | 21.97 m | 11.358% |
| WLTC | 21.45 m | 9.427% |

## 6. Conclusions and Prospect

This paper proposes a MSEHH-EV based on the planetary gear. Its dynamics, electrodynamics, electrical energy module, and hydraulic energy module are modeled mathematically. Based on the greatest recovery efficiency, the braking energy recovery control strategy is proposed in this paper, which is divided into small braking intensity, medium braking intensity, and emergency braking intensity according to the braking intensity, and the mechanical braking, as well as the switching of the recovery braking mode, is controlled according to the accumulator pressure and the battery soc threshold under various braking intensities. This paper establishes an MSEHH-EV simulation model based on AMESim,

and a joint simulation was established with Simulink through the joint simulation interface. The simulation was carried out under of UDDS, NEDC, and WLTC conditions respectively, and the simulation results were analyzed.

The simulation results are compared with the pure EV model in AMESim. It is concluded that the MSEHH-EV has a significant improvement in braking distance compared with the EV, with a reduction of 15.65 m, 21.97 m, and 21.45 m under the three operating conditions, respectively, which verifies that the model has better braking performance. In the comparison of the battery SOC, it was found that the simulation model consumes less power than the EV under all three operating conditions, including 17.766% less energy consumption under UDDS, 11.358% less energy consumption under NEDC, and 9.427% less energy consumption under WLTC, which proves that the hydraulic system in the model has a good energy-saving effect and has better range than the EV.

The MSEHH-EV based on the planetary gear has certain advantages in range and braking distance. It solves a series of problems such as low braking energy recovery efficiency, short driving range, and poor braking safety of EVs, which has important reference value for the future research and development of MSEHH-EVs.

This paper proposes the MSEHH-EV, and the simulation results show that its braking efficiency has a great advantage compared with pure EVs. The MSEHH-EV assembly has been completed, and a series of reliability tests and vehicle performance tests will follow. Finally, the braking test will verify the superiority of the MSEHH-EV braking efficiency.

**Author Contributions:** Conceptualization, J.W. and T.Z.; methodology, J.W.; software, J.W.; validation, H.Z. and J.W.; formal analysis, Z.Z.; investigation, Z.M.; resources, J.Y.; data curation, J.W.; writing—original draft preparation, J.W. All authors have read and agreed to the published version of the manuscript.

**Funding:** This research was funded by the National Natural Science Foundation of China, grant number 52075278, and the Municipal Livelihood Science and Technology Project of Qingdao, grant number 19-6-1-92-nsh.

**Conflicts of Interest:** The authors declare no conflict of interest.

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
