# Peer review of "Research on Braking Efficiency of Master-Slave Electro-Hydraulic Hybrid Electric Vehicle"

_electronics, doi:10.3390/electronics11121918_

Round 1
Reviewer 1 Report
The reviewer is satisfied with the innovative content of the paper and deems the research fit for this journal. Nevertheless, the paper is not suitable for publication in its present form due to inconsistency and poor presentation of the simulation results. The reviewer provides the following remarks to improve the quality of the paper:
~ Which vehicle category is being considered in this article ? Only on page 4 does the reader finally understand that the authors are writing about passenger cars.
~ Acronyms should be formulated in full at least once when they first appear in the text.
~ Which ECE regulation are the authors referring to on line 118 ?
~ It is not clear what type of Simulink model is being taken as a reference and under what assumptions. Is it experimentally validated ? How does it differ from the EV model ? The presentation of the simulation results should be dramatically improved.
~ Which paper are the authors referring to on line 399 ? Please, provide a reference.
~ The authors do not provide any description of Figure 7. In the reviewer's opinion, this is the core element of the simulation section. The reviewer understands that, based on the speed error, the driver's requested inputs are used to calculated the required powertrain torques. Please, provide a description for Figure 7.
~ It is not clear on what basis the proposed hybrid system should achieve improved braking performance, as the vehicle model follows a reference driving cycle. These type of simulations are generally used for efficiency analysis and not for performance analysis. To demonstrate better safety performance, the authors should consider braking tests where higher levels of deceleration are achieved.
Reviewer 2 Report
I found the paper very interesting, but there were many places where the wording made it difficult to follow. I've marked those issues on the attached pdf.

Reviewer 3 Report
As stated in the abstract "this paper proposes a master-slave electro-hydraulic hybrid drive system based on planetary row" which implies that the battery consumption rate is improved.
I find the topic interesting but I have some concerns about the paper.
1. Please define the meaning of acronyms. Example: Line 41 LSTM. I know that is long-short term memory but in a scientific paper, the acronyms are specified. Moreover, if you define EV, then you have to use EV in the rest of the manuscript. For example, line 42 the authors use electric vehicle when EV has already been defined previously.
2. I find the introduction a little long. The contributions of the paper are well explained.
3. I don't find any comparison between the authors' research and the results obtained. In section 5, we can only see the results obtained by the authors but not a comparison with current literature
4. How were the Simulink models validated to be sure that the results are reliable?
5. I'm not sure that the paper is well structured. I don't see a discussion section or a method section. I find many equations and theoretical concepts but I don't see that the method followed in this research is well explained.
I recommend reworking the paper and resubmitting it.
Reviewer 4 Report
The article is interesting and tecnologically novel. It´s well structurted and referenced.

Round 2
Reviewer 1 Report
In the reviewer's opinion, there is a lack of evidence that the proposed hybrid propulsion system can provide better braking performance based on the selected simulation scenario. In case the authors are unable to provide experimental and/or simulation results with this paper, the reviewer recommends mentioning in the final section the need for further research on the performance of the system.
Author Response
Point 1: In the reviewer's opinion, there is a lack of evidence that the proposed hybrid propulsion system can provide better braking performance based on the selected simulation scenario. In case the authors are unable to provide experimental and/or simulation results with this paper, the reviewer recommends mentioning in the final section the need for further research on the performance of the system
Response 1: Thanks for the suggestion raised by the reviewer. It has been modified in the original paper.
Reviewer 3 Report
The paper has been improved as well as its quality.
I recommend it for publication.
